# RCCT-ASPPNet: Dual-Encoder Remote Image Segmentation Based on Transformer and ASPP

Yazhou Li [1,2], Zhiyou Cheng [1,2], Chuanjian Wang [1,2,*], Jinling Zhao [1,2] and Linsheng Huang [1,2]

1 National Engineering Research Center for Analysis and Application of Agro-Ecological Big Data, Anhui University, Hefei 230601, China
2 School of Internet, Anhui University, Hefei 230039, China
* Correspondence: wcj_si@ahu.edu.cn

**Abstract:** Remote image semantic segmentation technology is one of the core research elements in the field of computer vision and has a wide range of applications in production life. Most remote image semantic segmentation methods are based on CNN. Recently, Transformer provided a view of long-distance dependencies in images. In this paper, we propose RCCT-ASPPNet, which includes the dual-encoder structure of Residual Multiscale Channel Cross-Fusion with Transformer (RCCT) and Atrous Spatial Pyramid Pooling (ASPP). RCCT uses Transformer to cross fuse global multiscale semantic information; the residual structure is then used to connect the inputs and outputs. ASPP based on CNN extracts contextual information of high-level semantics from different perspectives and uses Convolutional Block Attention Module (CBAM) to extract spatial and channel information, which will further improve the model segmentation ability. The experimental results show that the mIoU of our method is 94.14% and 61.30% on the datasets Farmland and AeroScapes, respectively, and that the mPA is 97.12% and 84.36%, respectively, both outperforming DeepLabV3+ and UCTransNet.

**Keywords:** remote image; deep learning; semantic segmentation; CNN; multiscale feature fusion; Transformer

## 1. Introduction

With the continuous development of artificial intelligence technology, computer vision has attracted much attention as one of the important research areas. Unlike the other fields of computer vision, such as image classification, object detection, and instance segmentation, current mainstream deep learning-based image semantic segmentation research aims to densely predict each pixel of an image using algorithms in which each pixel is labeled with its own category, thus achieving the goal of assigning semantic information to each identical pixel in the image [1]. The result of deep semantic segmentation gives computers a more detailed and accurate understanding of images and has a wide range of application needs in the fields of autonomous driving [2,3], face segmentation [4–7], and medical imaging [8–11].

Due to rapid advances in aerospace and sensor technology, it is easy and fast to obtain high-resolution satellite imagery and aerial imagery. Remote image semantic segmentation is one of the core contents of computer vision research [12–14]. With the active development of deep learning, remote image deep learning semantic segmentation networks have been continuously proposed, such as the FCN [15], UNet [16], SegNet [17], DeepLab [18–21], PSPNet [22], SETR [23], UCtransNet [24] models, etc. Some of these are based on convolutional neural network (CNN) and some are based on Transformer, which are explained below in terms of these two aspects.

### 1.1. Remote Image Segmentation Method Based on CNN

The CNN-based semantic segmentation method is one of the mainstream methods and mainly utilizes the encoder–decoder structure. The encoder typically uses convolutional neural networks and downsampling to reduce the resolution and to extract image feature

maps, while the decoder aims to transfer the low-resolution image and feature into image segmentation maps to achieve a pixel-level prediction, often using deconvolution [25] for upsampling, and the last layer of the network structure is mostly softmax classifiers to classify each pixel.

FCN replaces the fully connected layer at the end of the CNN with a convolutional layer and then upsamples it to obtain an image of the same size as the input. UNet also has an encoder–decoder structure (same as FCN), in which feature extraction is carried out in the first half and upsampling is carried out in the second half, and the skip connection layer in UNet merges low-level location information with deep-level semantic information. Similar to UNet, SegNet uses an encoder–decoder structure, but the encoder and decoder use different technologies. In addition, the encoder part of SegNet uses the first 13 layers of the VGG16 [26] convolutional network, each encoder layer corresponds to a decoder layer, and the output of the final decoder is fed into a softmax classifier to generate class probabilities for each pixel independently. DeepLabv1 [18] is based on two innovations: dilated convolution [27] and fully connected conditional random field. DeepLabv2 differs by proposing Atrous Spatial Pyramid Pooling (ASPP) [19], and DeepLabv3 [20] is based on further optimization of ASPP by adding convolution, BN operation, etc. DeepLabv3+ [21] is based on the structure of DeepLabV3 by adding an upsampling decoder module to optimize the accuracy of edges. These methods are widely used in remote image segmentation tasks and have obtained effective performance. However, the traditional CNN-based encoder–decoder network will lose some spatial resolution after a series of downsampling in the encoder stage, which affects the performance of semantic segmentation algorithms.

*1.2. Remote Image Segmentation Method Based on Transformer*

Transformer [28] was originally used in the field of natural language processing. Transformer is essentially an encoder–decoder structure. Transformer is based on the attention mechanism, which can solve the long-distance dependence problem. The attention mechanism has a better memory and can remember longer distance information. The most important thing is that attention supports parallelized computation, which is very suitable for remote images semantic segmentation. The transformer model is completely based on the attention mechanism, and it completely discards the structure of CNN.

Recently, some scholars used Transformer in semantic segmentation. Zheng et al. [23] proposed the SETR model for semantic segmentation, in which CNN is not used and so the resolution of the image is not degraded. Transformer cuts the image into multiple small pieces and encodes the ordering to achieve sequence-to-sequence encoding using attention mechanisms. Cao et al. [29] combined UNet with Transformer to extract multi-scale features. Although Transformer has achieved good performance in some semantic segmentation tasks, it has some limitations, such as larger model parameters and less segmentation capability than CNN. UCTransNet incorporates different feature layers of CNN into Transformer, which provides a new idea for multi-scale feature fusion. However, its CNN layer structure is simple, feature fusion is relatively single, the encoding and decoding methods are complex, and the ability to express various application scenarios of remote imaging is not sufficient.

*1.3. Remote Image Segmentation Method Based on CNN and Transformer*

To solve the problems caused by a single encoder, we concatenate CNN-based and Transformer-based network structures in order to compensate for the shortcomings of a single structure in remote image segmentation. First, we propose the Residual Multi-scale Channel Cross-Fusion with Transformer (RCCT) module as one of our encoding structures based on the multi-scale feature cross fusion approach of UCTransNet. Unlike UCTransNet, RCCT takes the first three feature layers of ResNet50 [30] as input and performs a cross-fusion of features, which capture the relationship between different feature layers in a Transformer way in order to obtain multi-scale semantic information. The output of RCCT is then concatenated into a whole feature layer and finally takes residual concatenation

with the three input feature layers. Second, To enhance the segmentation capability of the model on a remote image, the fourth feature layers of ResNet50 are input into the ASPP module, followed by connecting a Convolutional Block Attention Module (CBAM) [31] as the second encoding structure. The dual–encoder structure, which is called RCCT-ASPPNet, can effectively represent the global contextual information in the image and increase the receptive field, with a comprehensive performance higher than that of a single encoder.

### 1.4. Contributions

This paper addresses some challenges in the field of semantic segmentation of remote image by proposing a dual-encoder RCCT-ASPPNet. The main contributions of this paper can be illustrated in the following points.

First, an efficient remote image segmentation method based on CNN and Transformer is proposed. We design a Transformer-based RCCT structure. The first three feature layers of resnet50 are used as the input of RCCT, and the dependencies between each feature layer are learned in a Transformer cross-fusion manner. Then, we use the residual structure to link the fused input feature layer with the fused output feature layer.

Second, we not only extract features by transformer but also utilize the CNN-based ASPP module to obtain larger receptive field information, while adding channel attention and spatial attention after ASPP to learn deeper semantics. With the dual-encoder structures, we alleviate the problems of small targets, multiple scales, and diverse and complex categories in remote image.

Finally, we tested the method proposed on two datasets. The AeroScapes is a public dataset, which has a variety of perspectives, complex scenes, and more categories. The Farmland is a self-made dataset, which is top-down view, and the data have small objects. The experimental results show the effectiveness of our method. Our method has a further improvement in semantic segmentation of remote image with an mIoU of 94.14% and 61.30% on the datasets Farmland and AeroScapes [32] respectively.

### 1.5. Article Structure

In the Introduction section, an overview of deep learning, semantic segmentation techniques is provided, and the semantic segmentation method based on CNN and Transformer is introduced. The Methods and Data section explains the theoretical approach behind the model proposed in this paper and some parameter settings, in addition to the dataset used in this paper. The Results section mainly describes the ablation experiment and provides a comparison of different models. The Discussion section analyzes the experimental results, and their advantages and disadvantages. The Conclusions section summarizes the remote image semantic segmentation method proposed in this paper; in addition, the shortcomings of the method of this paper and the next step are explained.

## 2. Methods and Data

### 2.1. RCCT-ASPPNet Model Overview

Figure 1 shows a general description of the proposed approach. We use a two-layer encoder structure, including RCCT and ASPP-CBAM. The RCCT encoding module uses Transformer as the backbone network, and Transformer is used to cross fuse each feature layer to learn the feature relationship between the layers in an end-to-end way. The input and output are connected with residuals in a feature fusion way. The ASPP-CBAM encoding module combines ASPP with channel and spatial attention mechanisms, extracts feature maps of different receptive fields for concatenation, and then uses CBAM as an attention layer to learn the importance of channel and spatial importance.

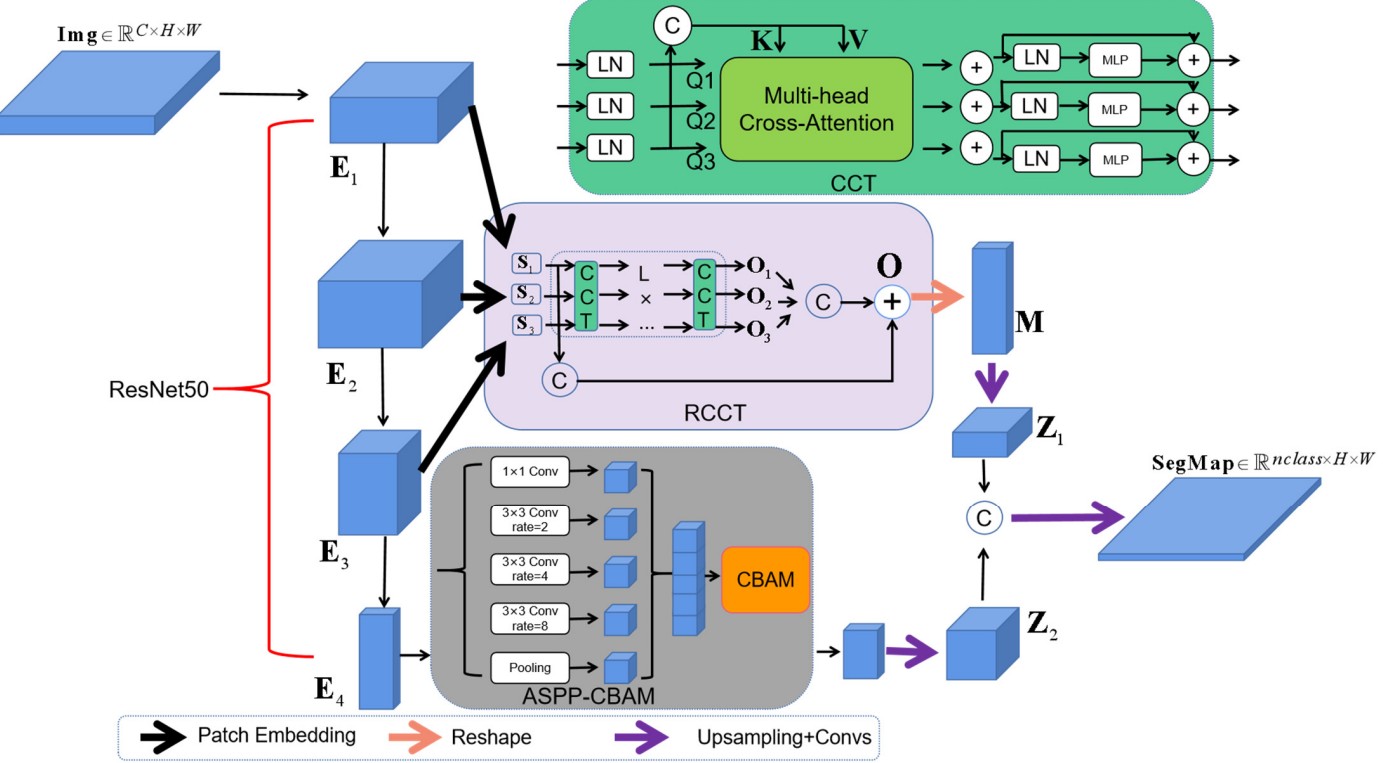

**Figure 1.** RCCT-ASPPNet model structure.

## 2.2. Residual Multi-Scale Channel Cross-Fusion Transformer (RCCT)

To address multi-scale feature fusion, we propose a RCCT module that takes advantage of long-dependency modeling in Transformer to fuse features from multi-scale encoders. The RCCT module has four steps: multi-scale feature embedding, multi-headed channel-based cross-attention, multi-layer perceptron (MLP), and residual feature fusion.

### 2.2.1. Multi-Scale Feature Embedding

Given as an input, the multi-scale feature embedding is the first three feature layers of ResNet50 $\mathbf{E}_i \in \mathbb{R}^{\frac{H \times W}{2^{2 \times (i+1)}} \times C_i}, (i = 1, 2, 3)$, where $C_i$ is the number of channel dimensions, and $C_1 = 256$, $C_2 = 512$, and $C_3 = 1024$. The standard Transformer accepts a sequence of token embeddings as input. To process 2D features, we reshape the feature $\mathbf{E}_i$ as a flattened block sequence $\mathbf{S}_i \in \mathbb{R}^{N \times C_i}, (i = 1, 2, 3)$, where $(\frac{H}{2^{i+1}}, \frac{W}{2^{i+1}})$ is the resolution of the original feature; $(p_i, p_i)$ is the resolution of each feature block; $p_1 = 16$, $p_2 = 8$, $p_3 = 4$; and $N$ (Equation (1)) is the number of feature blocks generated, that is, the effective input sequence length of the Transformer. In this process, we keep the channel size $N$ constant. Position embedding $\mathbf{S}_{pos} \in \mathbb{R}^{N \times C_i}$ is also added to the feature block to retain the spatial location information between the input feature blocks (Equation (2)).

$$N = \frac{\frac{H \times W}{2^{i+1} \times 2^{i+1}}}{p_i^2}, \tag{1}$$

$$\mathbf{S}_i = \mathbf{S}_i + \mathbf{S}_{pos}. \tag{2}$$

Then, we fuse the three embedded layers as the key and value (Equation (3)).

$$\mathbf{S}_{\sum} = Concat(\mathbf{S}_1, \mathbf{S}_2, \mathbf{S}_3). \tag{3}$$

2.2.2. Residual Channel Cross-Fusion Transformer

From Figure 1, we know that $\mathbf{S}_i$ is input into the multi-head channel cross attention module, followed by an MLP with a residual structure; the CCT module has been used L times; and we obtained the three outputs of the CCT module $\mathbf{O}_i$. Finally, the fused input feature layer is connected to the fused output feature layer in a residual manner, with $\mathbf{O}$ as the final output feature layer of RCCT. In this way, we learn the dependencies between the different input feature layers in a Transformer cross-fusion way.

As shown in Figure 2, the multi-head cross attention module contains four inputs, that is, three embedded layers as the query matrix and one integrated embedded layer $\mathbf{S}_\Sigma$ as the key $\mathbf{K}$ and the value $\mathbf{V}$.

$$\mathbf{Q}_i = \delta(\mathbf{S}_i)W_{\mathbf{Q}_i}, \mathbf{K} = \mathbf{S}\sum W_{\mathbf{K}}, \mathbf{V} = \mathbf{S}\sum W_{\mathbf{V}}, \tag{4}$$

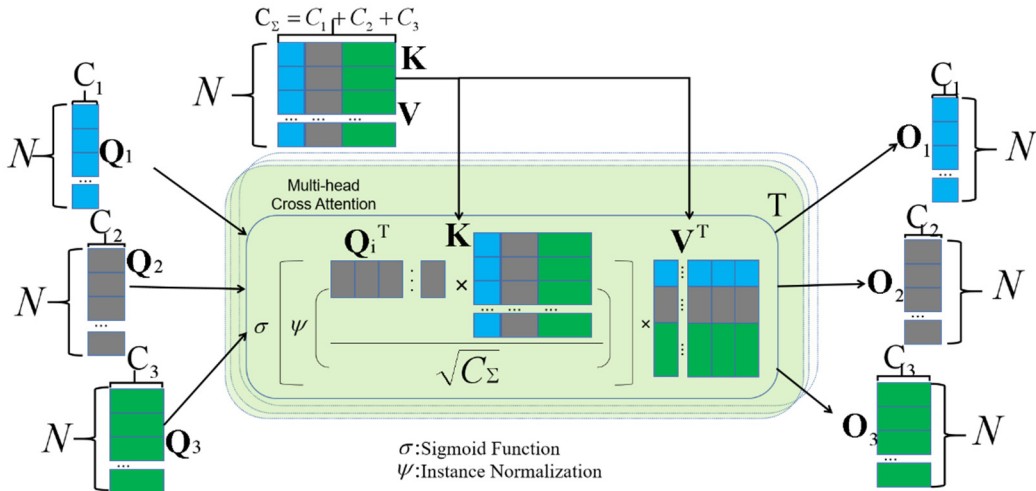

**Figure 2.** Multi-head cross attention.

$W_{\mathbf{Q}_i} \in \mathbb{R}^{C_i \times C_i}$, $W_{\mathbf{K}} \in \mathbb{R}^{C_\Sigma \times C_\Sigma}$, and $W_{\mathbf{V}} \in \mathbb{R}^{C_\Sigma \times C_\Sigma}$ are different input weights and $\delta(\cdot)$ represents layer normalization. Meanwhile, $\mathbf{Q}_i \in \mathbb{R}^{N \times C_i}$, $\mathbf{K} \in \mathbb{R}^{N \times C_\Sigma}$, and $\mathbf{V} \in \mathbb{R}^{N \times C_\Sigma}$. The formula for cross attention is described as follows:

$$Attention(\mathbf{Q}_i, \mathbf{K}, \mathbf{V}) = \left\{ \sigma\left[\psi\left(\frac{\mathbf{Q}_i^T \mathbf{K}}{\sqrt{C_\Sigma}}\right)\right] \mathbf{V}^T \right\}^T, \tag{5}$$

$\psi(\cdot)$ and $\sigma(\cdot)$ represent the instance normalization and softmax function, respectively. We adopt instance normalization, which can normalize each instance matrix of multi-head attention so that the gradient can spread smoothly. $Attention(\mathbf{Q}_i, \mathbf{K}, \mathbf{V}) \in \mathbb{R}^{N \times C_i}$ is the same size as the input $\mathbf{Q}_i$. Because we have $H_n$ heads' attention, the output result after multi-head cross attention is calculated as follows:

$$MHAttention_i = \left[ \begin{array}{c} Attention(\mathbf{Q}_i^1, \mathbf{W}, \mathbf{K}) + Attention(\mathbf{Q}_i^2, \mathbf{W}, \mathbf{K}) \\ +, \ldots, + Attention(\mathbf{Q}_i^{H_n}, \mathbf{W}, \mathbf{K}) \end{array} \right] / H_n, \tag{6}$$

where $H_n$ is the number of heads. Then, combining MLP and residual operation, the output is as follows:

$$\mathbf{O}_i = \delta(MHAttention_i) + MLP(\mathbf{Q}_i + MHAttention_i), \tag{7}$$

The operation in Equation (7) is repeated L times to establish an L-layer Transformer, in which the output of the L-layer is $\mathbf{O}_i \in \mathbb{R}^{C_i \times N}$. In this paper, $H_n$ and L are both set to 4. The output of the last layer is a multi-scale residual fusion, and the formula is as follows:

$$\mathbf{O} = Concat(\mathbf{S}_1, \mathbf{S}_2, \mathbf{S}_3) + Concat(\mathbf{O}_1, \mathbf{O}_2, \mathbf{O}_3), \tag{8}$$

$\mathbf{O} \in \mathbb{R}^{C_\Sigma \times N}$ is the final output of the RCCT module.

Matrix $\mathbf{O}$ obtains the feature layer $\mathbf{M} \in \mathbb{R}^{C_\Sigma \times \sqrt{N} \times \sqrt{N}}$ through the reshape operation, and we obtain $\mathbf{Z}_1$ through upsampling and the convolutional operation.

### 2.3. CBAM Module

In the output feature map of the ASPP module, CBAM, as shown in Figure 3, infers the attention map in turn along two independent dimensions (channel and spatial) and then multiplies the attention map and the input feature map for adaptive feature optimization. Now, the channel and spatial attention modules are distinguished.

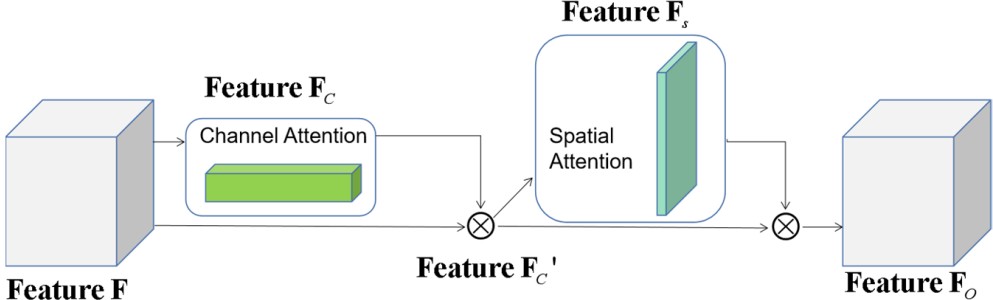

**Figure 3.** CBAM sketch map.

Figure 4 shows the structure of the channel attention module CA. It models the importance of each feature channel and then enhances or suppresses different channels. The output $\mathbf{F}$ of ASPP is used as the input of channel attention. $\mathbf{F}$ goes through the maximum pooling layer and the average pooling layer. The two outputs are connected to the same MLP, and their parameters are shared. The two outputs of MLP are added, and a sigmoid function is finally connected to obtain $\mathbf{F}_C$.

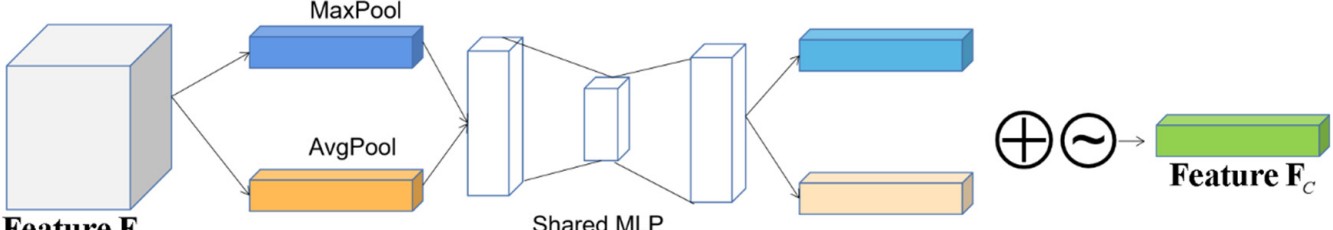

**Figure 4.** Channel attention module.

Figure 5 depicts the structure of the SA module. Not all regions in the image contribute equally to the task. Only the regions related to the task need to be concerned. The SA model aims to find the most important part of the network for processing. $\mathbf{F}_C{}'$ is the input of SA, $\mathbf{F}_C{}'$ also goes through the maximum pooling layer and the average pooling layer, and their outputs are concatenated on the channel dimension. Then, a convolution layer and a sigmoid function are connected to obtain feature $\mathbf{F}_S$.

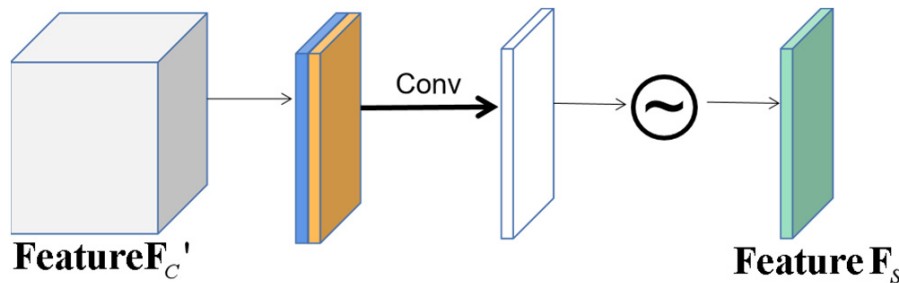

**Figure 5.** Spatial attention module.

*2.4. Dual Encoders of ASPP-CBAM and RCCT*

ASPP samples the given input in parallel with a convolution of different dilation rates, which is equivalent to capturing the context of the image at multiple scales. The last feature layer $\mathbf{E}_4 \in \mathbb{R}^{\frac{H \times W}{2^{2 \times (4+1)}} \times C_4}$ of ResNet50 is used as the input of ASPP. $\mathbf{E}_4$ went through four different convolution operations with different dilation rates to extract the features under different receptive fields. ASPP includes a $1 \times 1$ convolution layer and three $3 \times 3$ dilated convolution, dilated rate = {2, 4, 8}, and pooling layers, as shown in Figure 1. The five feature layers are merged as the output of ASPP and enter the CBAM attention layer. Then, after the upsampling and convolution operations, we obtain the ASPP-CBAM output layer $\mathbf{Z}_2$. Finally, the output layer is merged with the output layer of RCCT $\mathbf{Z}_1$ by the following equation:

$$\mathbf{Z} = Concat(\mathbf{Z}_1, \mathbf{Z}_2), \tag{9}$$

*2.5. Data*

2.5.1. Self-Made Dataset

The data come from the agricultural remote sensing image (Farmland) taken by UAV, as shown in Figure 6, which is divided into six categories: grassland, construction land, cultivated land, forest land, garden land, and other lands. The image is from an overhead perspective, and the difference between target sizes is large. The drone is a DJI M300 RTK with a flight altitude of 5 km, and a DJI P1 camera with an image size of 8192 × 5460 pixels and a sensor size of 35.9 mm × 24 mm, with 45 million effective pixels and an image element size of 4.4µm. In addition, the data were collected in November 2021.

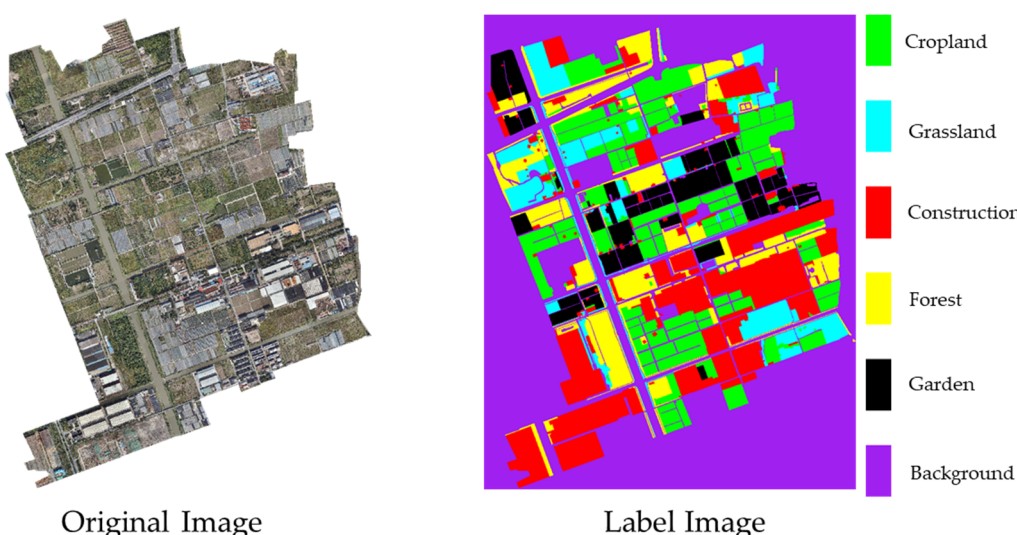

Original Image　　　　　　　　　　　　　Label Image

**Figure 6.** Farmland including raw and labeled images.

To enhance the generalization ability of the model, this study adopts the method of data enhancement. The original remote images and labeled images were first generated by

cropping multiple images of size 512 × 512 pixels, expanding the data set using random rotation, adding noise, flipping, and using other data enhancement methods, as shown in Figure 7. A total of 5000 images were generated, which were then divided into the training and validation sets according to a 4:1 ratio.

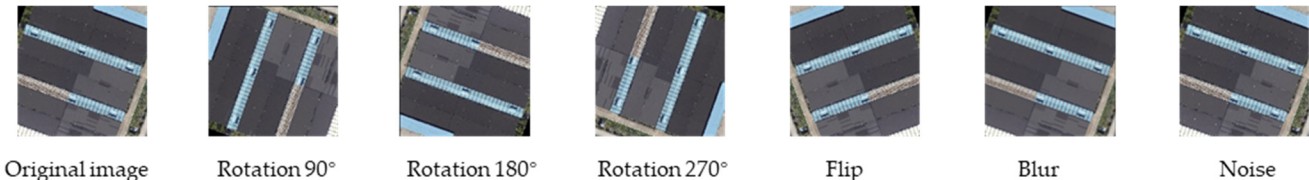

**Figure 7.** Data enhancement methods.

### 2.5.2. AeroScapes Dataset

The AeroScapes semantic segmentation dataset includes images captured from 5 m to 50 m height using commercial UAVs. This dataset provides 3269 720 pixel × 1280 pixel resolution images and real land surface labels for 12 classes. Figure 8 shows the information of 11 categories and background categories. The dataset has a variety of perspectives, target scales vary greatly, and there are many small targets.

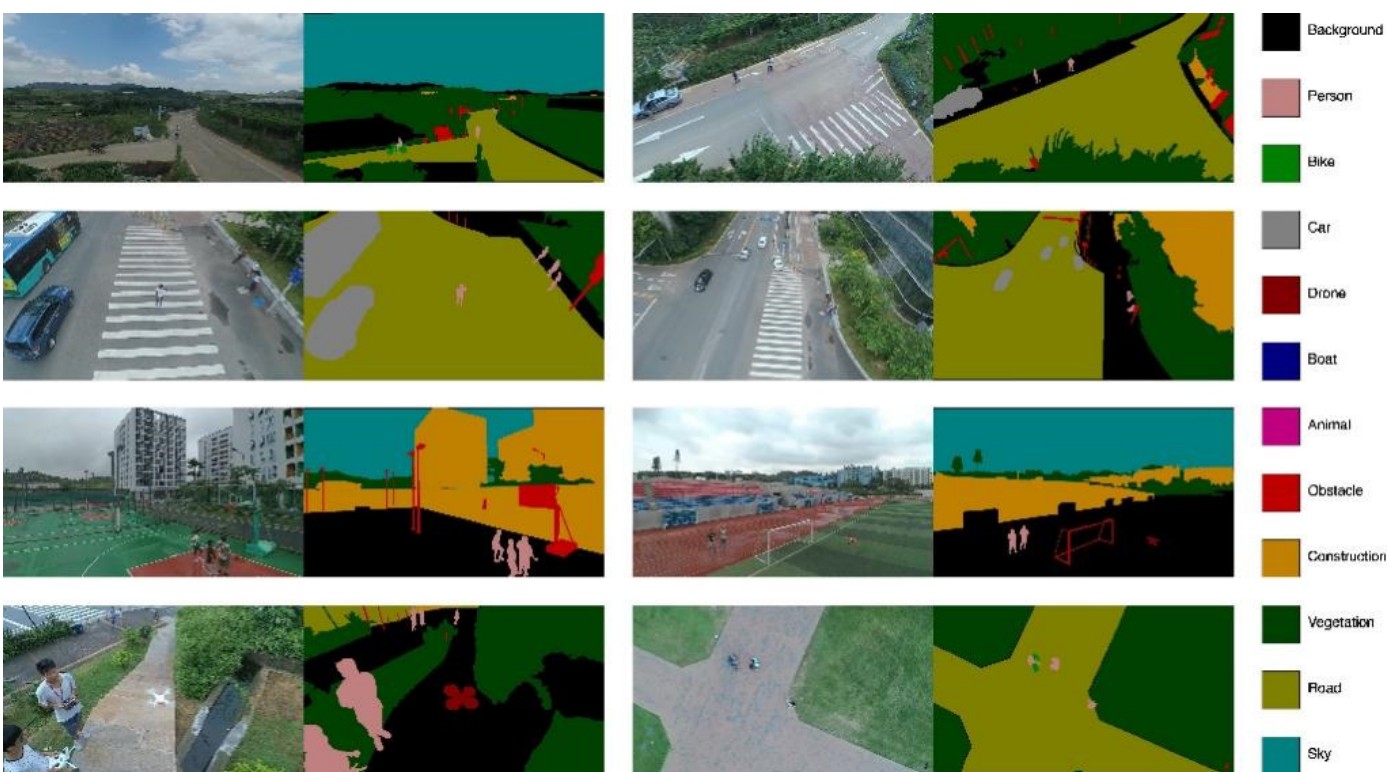

**Figure 8.** AeroScapes dataset.

## 3. Results

### 3.1. Experimental Environment and Parameter Setting

We use Python as the deep learning framework, JetBrains PyCharm 2021 as the development platform, and Python 3.8 as the development language. All models are trained and tested on computers configured with Intel Core (TM) i7-10700K CPUs and NVIDIA GeForce GTX 3090 Ti graphics cards.

The model uses a poly [33] strategy to reduce the learning rate. The formula is as follows:

$$lr = initial\_lr \times (1 - \frac{epo}{maxepo})^{power}, \tag{10}$$

The initial learning rate *initial_lr* is set to 0.001, and *power* is set to 0.9. The maximum number of iterations *maxepo* is 300, and *epo* represents the current number of iterations. In this study, the batch size is set to 8, and the Stochastic gradient descent SGD [34] optimizer is used to optimize the poly algorithm and network parameters. The model's backbone network Resnet50 uses the trained weight of the dataset ImageNet [35] as the initial weight.

### 3.2. Evaluation Index and Loss Function

To quantify the effect of the evaluation model, this study uses the most common evaluation index in the field of semantic segmentation: mean Intersection over Union (*mIoU*).

$$mIoU = \frac{1}{c+1} \sum_{i=0}^{c} \frac{p_{ii}}{\sum_{j=0}^{c} p_{ij} + \sum_{j=0}^{c} p_{ij} - p_{ii}}, \tag{11}$$

where $p_{ii}$ represents the number of pixels predicted by category $i$ to category $i$; $c + 1$ is the total number of categories; and $p_{ij}$ is the number of pixels predicted by category $i$ to category $j$.

We also use mean Pixel Accuracy (*mPA*) as another evaluation index.

$$mPA = \frac{1}{c+1} \sum_{i=0}^{c} \frac{p_{ii}}{\sum_{j=0}^{c} p_{ij}}, \tag{12}$$

The loss function of Lovasz Softmax [36] is usually used to evaluate semantic segmentation using the Jaccard index [37], also known as the IoU index. In Equation (13), $y*$ is the true label, and $y$ is the predicted value. The Jaccard index of category $c$ is defined as follows:

$$J_c(y*, y) = \frac{|\{y* = c\} \cap \{y = c\}|}{|\{y* = c\} \cup \{y = c\}|}, \tag{13}$$

The corresponding loss function is as follows:

$$L_{J_c}(y*, y) = 1 - J_c(y*, y), \tag{14}$$

Figure 9 shows the trend of the training set and validation set of the proposed method on the Farmland dataset with the number of iterations. The curve decreases more rapidly in the first 50 iterations of the loss value and stabilizes after the number of iterations exceeds 250.

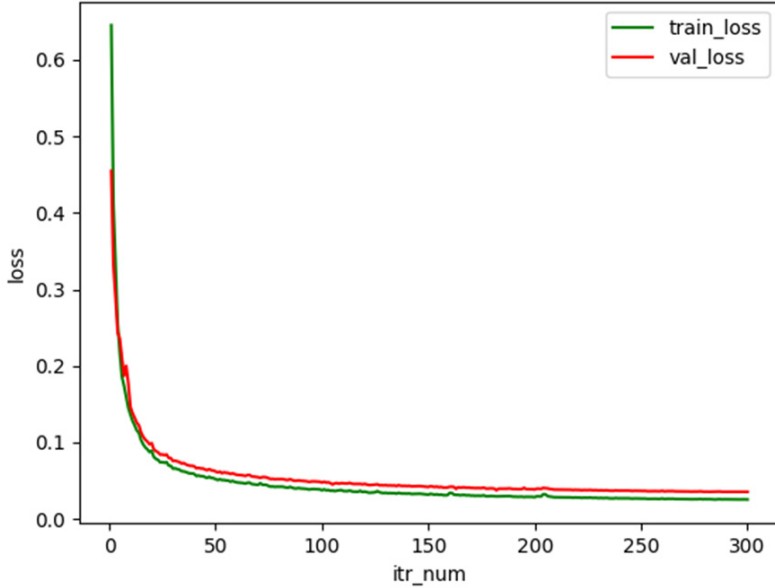

**Figure 9.** Lovasz Softmax loss function change curve.

### 3.3. Ablation Experiment of RCCT Module with Different Feature Combinations

To verify the influence of input $\mathbf{E}_i$ of different scales on the model, four groups of RCCT module inputs with different combinations are designed for experimental comparison. The four groups of experiments were combined with the ASPP module.

The following conclusions can be drawn through the comparison of four groups of experiments in Table 1:

**Table 1.** Comparison of different feature combination inputs of the RCCT module.

| Input Feature | Farmland | | AeroScapes | |
|:---:|:---:|:---:|:---:|:---:|
| | mIoU (%) | mPA (%) | mIoU (%) | mPA (%) |
| $\mathbf{E}_1$, $\mathbf{E}_2$, $\mathbf{E}_3$ | 93.97 | 96.83 | 60.86 | 84.20 |
| $\mathbf{E}_1$, $\mathbf{E}_2$ | 93.46 | 96.75 | 60.47 | 83.61 |
| $\mathbf{E}_1$, $\mathbf{E}_3$ | **94.11** | **97.10** | **61.19** | **84.58** |
| $\mathbf{E}_2$, $\mathbf{E}_3$ | 94.00 | 96.97 | 61.05 | 84.55 |

Not all feature inputs can increase the performance of the model, and some features can reduce the accuracy of the model. For example, the mIoU and mPA of the combination $\mathbf{E}_1$, $\mathbf{E}_2$, and $\mathbf{E}_3$ are lower than that of combination $\mathbf{E}_1$ and $\mathbf{E}_3$ and the combination $\mathbf{E}_2$ and $\mathbf{E}_3$, indicating that, when feature input $\mathbf{E}_1$ and $\mathbf{E}_2$ are used together, they will have negative effects. Considering that the combination effect of $\mathbf{E}_1$ and $\mathbf{E}_3$ is better than that of $\mathbf{E}_2$ and $\mathbf{E}_3$, it indicates that $\mathbf{E}_1$ and $\mathbf{E}_3$ are the optimal combination, that is, the effect of $\mathbf{E}_1$ is greater than that of $\mathbf{E}_2$. Therefore, $\mathbf{E}_2$ needs to be removed from the final RCCT module feature input, thus improving the model performance and reducing the model parameters.

A high-level feature is necessary. From the experimental results, combination $\mathbf{E}_1$ and $\mathbf{E}_2$ has the worst effect. As long as $\mathbf{E}_3$ appears in the combination, its mIoU will be 0.5% or higher than that without $\mathbf{E}_3$. In the combination with $\mathbf{E}_3$, mIoU has little difference. Therefore, high-level feature $\mathbf{E}_3$ plays a crucial role in the model.

### 3.4. Ablation Experiment of Different Attention Combinations in the ASPP Module

The combination of the ASPP module with SA and CA can effectively increase the mIoU and mPA of the model. We have verified the effect of different attentions on the module (Table 2). The input combination of the RCCT module in the experiment is $\mathbf{E}_1$, $\mathbf{E}_2$, and $\mathbf{E}_3$.

**Table 2.** ASPP module: different attention combinations.

| Attention Combination | Farmland | | AeroScapes | |
|:---:|:---:|:---:|:---:|:---:|
| | mIoU (%) | mPA (%) | mIoU (%) | mPA (%) |
| ASPP | 93.97 | 96.83 | 60.86 | 84.20 |
| ASPP + CA | 94.12 | 97.06 | 61.22 | 84.22 |
| ASPP+ SA | 94.02 | 96.87 | 61.13 | 84.25 |
| ASPP + CBAM | **94.14** | **97.12** | **61.30** | **84.36** |

From the experiment results in Table 2, the CA and SA modules are more effective than the ASPP without the attention module. The mIoU and mPA values are higher when both attention modules act on ASPP simultaneously than when one attention module is used alone. Therefore, CBAM can effectively improve the performance of ASPP by paying attention to the feature information under different visual fields in space and channel.

### 3.5. Ablation Experiment of Dual Encoders

To compare the effects of the two encoders on the model, experiments using RCCT module alone and ASPP module alone are designed. Table 3 shows the results. The RCCT used an $\mathbf{E}_1$, $\mathbf{E}_2$, and $\mathbf{E}_3$ input combination in the experiment, and the ASPP did not use the CBAM module.

**Table 3.** Comparison of two encoder modules.

| Module | Farmland | | AeroScapes | |
|---|---|---|---|---|
| | mIoU (%) | mPA (%) | mIoU (%) | Mpa (%) |
| RCCT | 91.52 | 93.68 | 59.72 | 82.18 |
| ASPP | 92.72 | 94.81 | 60.35 | 80.65 |
| RCCT + ASPP | **93.97** | **96.83** | **60.86** | **84.20** |

Table 3 shows the experimental results of the two independent encoders. The accuracy of the dual encoding structure of RCCT and ASPP is greater than that of either encoding structure, illustrating that both Transformer-based multiscale feature fusion and ASPP with different dilation rates are important components of semantic segmentation of remote images. From another aspect, the accuracy of the Transformer-based RCCT module alone is lower than that of the ASPP module in Farmland, reflecting that ASPP is more capable than RCCT at overhead angle semantic segmentation tasks. Moreover, the difference in mIoU between RCCT and ASPP on AeroScapes is small, but for mPA, RCCT is better than ASPP, so RCCT performs better on multi-angle and more categories datasets.

### 3.6. Comparative Experiment of Different Network Models

We compared our method with some mainstream remote image semantic segmentation methods, including FCN-8s, UNet, DeepLabV3+, SETR, and UCTransNet. Table 4 shows the experimental results of various models in our dataset and AeroScapes dataset.

**Table 4.** Comparison of different networks.

| Network Model | Farmland | | AeroScapes | | Model Parameters (M) |
|---|---|---|---|---|---|
| | mIoU (%) | mPA (%) | mIoU (%) | mPA (%) | |
| FCN-8s | 92.21 | 96.43 | 40.23 | 78.69 | **80** |
| UNet | 89.38 | 95.06 | 42.38 | 50.41 | 124 |
| DeepLabV3+ | 92.80 | 96.28 | 59.63 | 67.07 | 170 |
| SETR | 49.53 | 64.82 | 30.63 | 37.38 | 348 |
| UCTransNet | 92.82 | 93.27 | 52.33 | 81.67 | 363 |
| RCCT-ASPPNet | **94.14** | **97.12** | **61.30** | **84.36** | 411 |

Table 4 shows that the mIoU of DeepLabV3+ is 92.80% and 59.63% in Farmland and AeroScapes datasets, respectively, which is the best performance among the CNN-based network models. However, the mPA of FCN-8s is 78.69% in AeroScapes datasets, and it is better than DeepLabV3+, but the mIoU of FCN-8s is only 40.23%. Moreover, UCTransNet performs best among Transformer-based models. Our RCCT-ASPPNet network model outperforms the other models with mIoU of 94.14% and 61.30% and mPA of 97.12% and 84.36% for the Farmland and AeroScapes datasets, respectively, which is a dual-encoder structure based on CNN and Transformer.

## 4. Discussion

### 4.1. Visual Analysis

Figure 10 shows the effect of Farmland data prediction. The performance of CNN-based FCN-8s, UNet, and DeepLabV3+ is evidently better than that of Transformer-based SETR and UCTransNet. RCCT-ASPPNet combines CNN and Transformer, and its performance is relatively good in Farmland. Figure 11 shows that the prediction results of FCN-8s, UNet, DeepLabV3+, and SETR are relatively poor in AeroScapes. UCTransNet and RCCT-ASPPNet have relatively good prediction effects in (l). Therefore, the improved Transformer model is better than the traditional model. Although RCCT-ASPPNet showed some misdividing in (k), overall, the segmentation effect of RCCT-ASPPNet in different views and small targets was better than UCTransNet. By comparing the three lines of

pictures in (g), (h), and (i), our model handles the best details in terms of the prediction results under people's tilt angle of view and top angle.

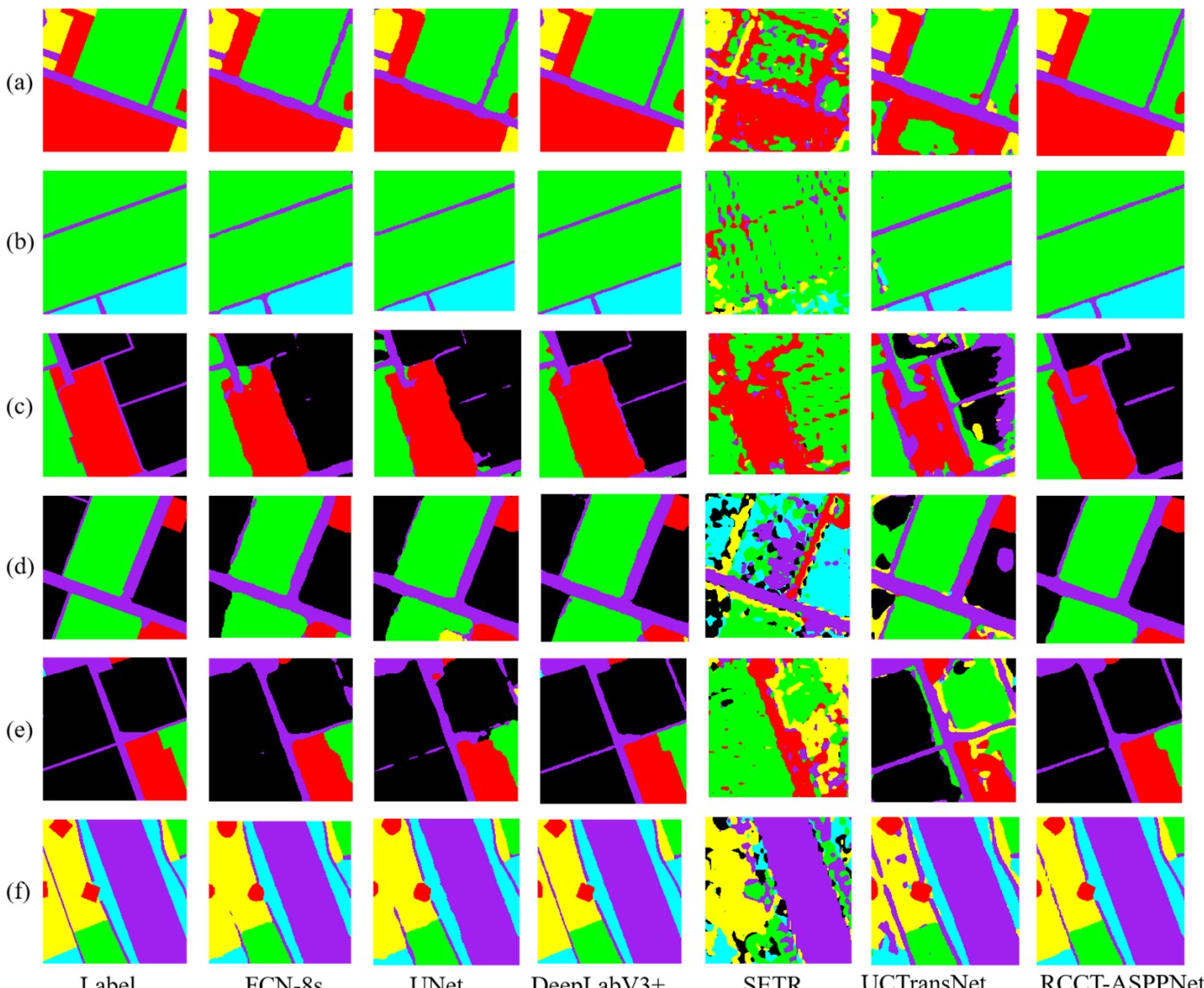

**Figure 10.** Farmland data prediction results, from left to right, are real labels, FCN, UNet, Deeo-LabV3+, SETR, UCTransNet, and RCCT-ASPPNet, from top to bottom, (**a**–**f**) are the selected 6 sets of Farmland test images.

Combining the performance of the above two datasets, the CNN-based model is more effective at processing the top-view images in the Farmland dataset. However, for the multiple views and more categories in the AeroScapes dataset, the CNN model does not perform well. Moreover, UCTransNet performs the opposite. The combination of Transformer and ASPP can compensate for the shortcomings of each.

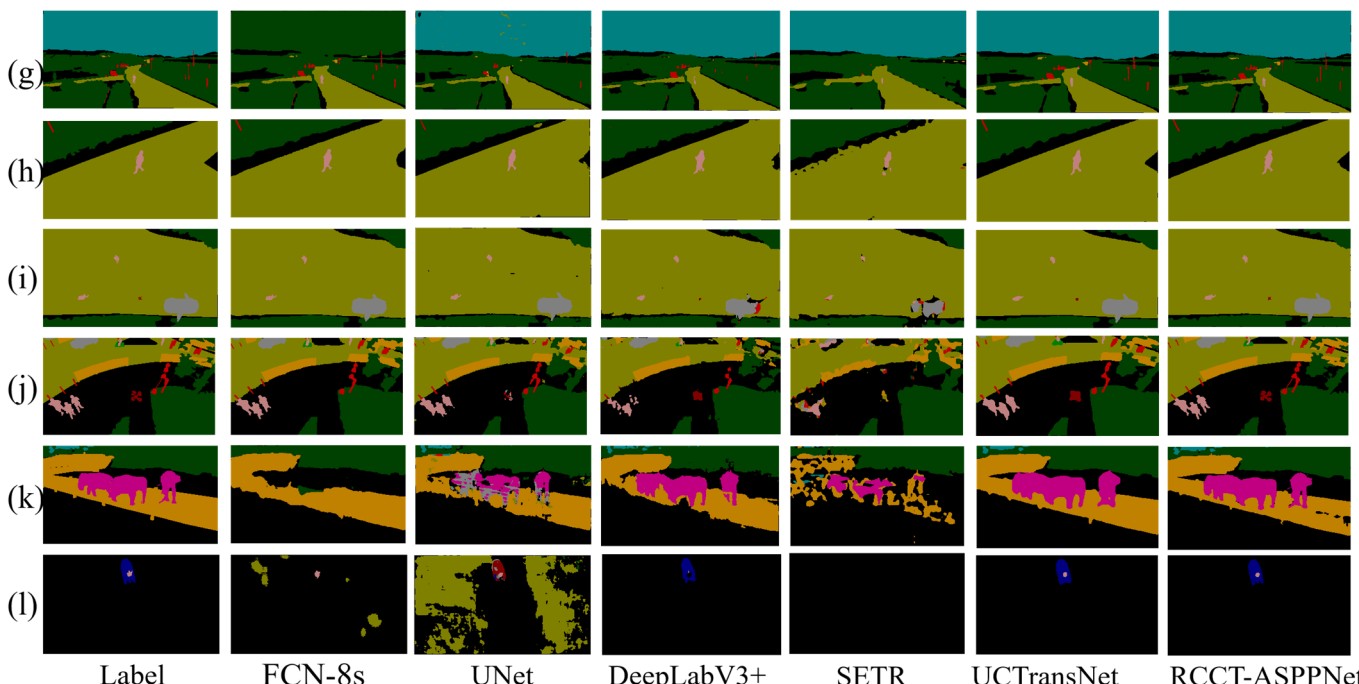

**Figure 11.** AeroScapes data prediction results, from left to right, are real labels, FCN, UNet, Deeo-LabV3+, SETR, UCTransNet, and RCCT-ASPPNet, from top to bottom, (**g–l**) are the selected 6 sets of AeroScapes test images.

### 4.2. Analysis of Experimental Results

#### 4.2.1. Analysis between Network Models

The performance of the six models on Farmland and AeroScapes is compared, as presented in Table 4. Among all the models, SETR has the worst results on both datasets because the simple Transformer model does not combine multiscale features and loses many low-level semantic features. In addition, the SETR model is more homogeneous and difficult to segment remote images in complex situations. The FCN-8s network model uses multiscale feature fusion, which can effectively learn multiple features while up-sampling using deconvolution, preserving the spatial information of the original input image. However, FCN-8s performs poorly in AeroScapes images with multiple categories and views, with only 40.23% mIoU. The reason is that FCN-8s has difficulty extracting higher-level features and the multiscale fusion approach is simply concatenated, which does not sufficiently learn the relationships among the feature layers. The ASPP module of DeepLabV3+ convolution with higher-level features at different dilate rates extracts the feature information under different fields of view and fuses the lower-level feature layers. The mIoU of DeepLabV3+ on Farmland and AeroScapes is 92.80% and 59.63%, respectively, which achieved good results but at the same time ignored the cross information between feature layers; the mPA of DeepLabV3+ on AeroScapes is only 67.07%. UCTransNet used Transformer to cross-fuse all input feature layers, which solved the problem of insufficient feature fusion, and it has a mPA metric of 81.67% on dataset AeroScapes, which exceeds most network models; this shows that UCTransNet has better performance in handling multi-category and multi-angle datasets. However, the different fields of view of feature layers are equally important; therefore, UCTransNet does not perform as well as the CNN-based model on Farmland. RCCT-ASPPNet cross-fuses some input features and then uses the residual method to connect the front and back feature layers as an encoder. Meanwhile, RCCT-ASPPNet uses the ASPP module to process the high-level features and CBAM to learn the channel and spatial information, which is the second encoder. RCCT-ASPPNet considers the feature fusion method and the field of view information of the feature layer to achieve optimal results on both Farmland and AeroScapes.

In this paper, a dual encoder model was proposed. The first encoder is used to cross-fuse the first three feature layers of ResNet50 using Transformer to learn multi-scale information; it can learn the dependencies of a feature layer with other feature layers. In addition, we added a residual module before and after fusion to prevent gradient disappearance; the second encoder module uses ASPP to process the highest feature layer of ResNet50 to obtain a larger receptive field and uses CBAM to learn its channel attention and spatial attention. In this paper, we used the common evaluation metrics of semantic segmentation, mIoU and mPA, to measure the accuracy of the model. RCCT-ASPPNet outperforms other semantic segmentation models on both Farmland and Aeroscapes in Table 4; in addition, we can see from Figures 10 and 11 that the algorithm in this paper has better segmentation performance in handling small targets and multi-scale objects and using one object in multiple views.

### 4.2.2. Analysis of Ablation Experiments

By introducing two encoders, the RCCT module and the ASPP module, the experimental design in Table 3 shows that the effect of double coding is more accurate than that of single coding. In addition, this study was designed for the influence of different input feature combinations of RCCT modules on the experimental results. Table 1 shows that the feature combination of $E_1$ and $E_3$ is optimal, and the mIoU and mPA on Farmland reach 94.11% and 97.10%, while, 61.19% and 84.58%, respectively, on AeroScapes, so not all features are effective combinations. From another aspect, the experimental results show that the $E_3$ feature is essential. The experimental design in Table 2 can obtain the impact of different attention mechanisms on the ASPP module. When the attention mechanism is not applicable, mIoU and mPA are 93.97% and 96.83% of the lowest value on Farmland, the same for AeroScapes. When CA or SA is used alone, mIoU and mPA show slight increases, whereas when CBAM is used, the index reaches the highest value. Therefore, CBAM has a certain effect on the ASPP module.

## 5. Conclusions

In this work, we proposed an effective RCCT-ASPPNet network model for the semantic segmentation of remote image. We used a dual-encoder structure, including a residual multiscale channel cross-fusion Transformer to address multiscale feature fusion and ASPP to address information extraction at different scales on a single feature layer. Extensive experiments evaluated that the proposed model can effectively alleviate the problems of remote images with large-scale variations, small target objects, and diverse viewpoints. RCCT-ASPPNet outperforms the CNN-based DeepLabV3+ and Transformer-based UC-TransNet. Compared with other state-of-the-art remote image semantic segmentation methods, RCCT-ASPPNet's accuracy has a first-class performance.

Although our experimental results have achieved good results, the effects on other data sets are unclear, so we will study the performance of this algorithm on each data set later. From another aspect, Table 4 shows that our model parameters are larger than those of other algorithms, which is very unfriendly for the real-time segmentation. Therefore, future work should balance the accuracy and efficiency of the model.

**Author Contributions:** Conceptualization, Y.L., Z.C. and C.W.; methodology, Y.L., Z.C. and C.W.; formal analysis, C.W. and J.Z.; data curation, C.W.; writing—original draft preparation, Y.L.; writing—review and editing, Z.C., J.Z. and L.H.; visualization, L.H.; supervision, Z.C. and C.W.; funding acquisition, C.W. and L.H. All authors have read and agreed to the published version of the manuscript.

**Funding:** This research was funded by Natural Science Foundation of China, grant number 31971789, and Excellent Scientific Research and Innovation Team (2022AH010005), and National Key Research and Development Project, grant number 2017YFB050420.

**Data Availability Statement:** https://github.com/ishann/aeroscapes (accessed on 26 November 2022); https://pan.baidu.com/s/1wK4qCwqfMOTec2bI7JrMmA?pwd=abcd (accessed on 26 November 2022).

**Acknowledgments:** We thank all editors and reviewers for their valuable comments and suggestions, which improved this manuscript.

**Conflicts of Interest:** The authors declare no conflict of interest.

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
