# Peer review of "RCCT-ASPPNet: Dual-Encoder Remote Image Segmentation Based on Transformer and ASPP"

_remotesensing, doi:10.3390/rs15020379_

Round 1
Reviewer 1 Report
This manuscript introduces a method of UAV Image Segmentation Based on Transformer and ASPP .In summary, the research is interesting, but the current document has several weaknesses that must be strengthened in order to obtain a documentary result that is equal to the value of the publication.
(1) To carry out a research work, it is necessary to have sufficient understanding and learning of the related field. However, this document only contains 28 papers, which seems difficult to provide sufficient theoretical support for this study.
(2) The first paragraph introducing the research topic may present a much broad and comprehensive view of the problems related to your topic with citations to authority references (Recognition and localization methods for vision-based fruit picking robots: a review. Frontiers in Plant Science 2020, 11:510.).
(3) On the general level, the abstract summarizes the main work content of this paper, but does not reflect the innovation of this work. At the same time, because the existing description of this part is too general, it does not reflect the key issues solved by this research.
(4) In the introduction section mentioned a kind of UAV Image Segmentation method realized by the Transformer and ASPP. However, there is no description of the application scenario of this research, nor a comprehensive analysis of the dilemma of semantic segmentation in specific scenarios. Therefore, the purpose and significance of this study are not specific and targeted enough to reflect the necessity of this study.
(5) This paper lacks both a comprehensive analysis of the semantic segmentation network and a summary of existing encoder and decoder work, which is insufficient to highlight the advantages of this study.
(6) The novelty of the study is not apparent enough. In the introduction section,please highlight the contribution of your work by placing it in context with the work that has done previously in the same domain. And,there should also be a clearer description of the structure of the article in this section.
(7) Vision technology integrated with deep learning is emerging these years in various engineering fields. The authors may add more state-of-art articles for the integrity of the introduction. For object detection, please refer to A Study on Long–Close Distance Coordination Control Strategy for Litchi Picking; Fruit detection and positioning technology for a Camellia oleifera C. Abel orchard based on improved YOLOv4-tiny model and binocular stereo vision.
(8) Based on the contents developed in chapter 2.5.1, it is noted that there are obvious errors in the description of parameter settings. In addition, the description of datasets in Section 2.6.2 also has obvious errors.
(9) As described in Section 2.5.1, this study uses only one index to evaluate the model effect, and lacks evaluation indexes for accuracy, which is not a scientific evaluation method.
(10) In Section 2.6.1, the sampling situation of the Self-Made Dataset should be described in more detail, including the sampling time of the dataset, parameters of image acquisition equipment and other information. Similarly, there is no explanation of the color and label of the Label Image.
(11) In chapter 4 - Results, due to the lack of accurate evaluation index, the verification content of the algorithm performance is not sufficient and comprehensive, and it is necessary to carry out various and deeper tests and analyses.
(12) For ablation tests and discussion of the model, performance verification should be performed on both datasets.
(13) In this paper, it is mentioned that the loss function of the model in section 2.5.1. However, the loss curves of the training set and the validation set are not shown and analyzed in the conclusion.
(14) The positions of figure 9, and figure 10 seem to be more appropriate in Section 3. At the same time, the phenomenon analyses from Section 3 are not deep enough, which should be reflected more comprehensively.
(15) In Section 4.3, it is mentioned in the defect description of the algorithm that the parameters of the proposed model are larger than those of other algorithms, and the comparison should be presented in a clearer way such as a table.
(16) In conclusion part, it should mention the scope for further research as well as the implications/application of the study.
Reviewer 2 Report
[1. Introduction]
n Line 67: The sentence looks awkward because the clause "Attention module" is repeated.
n Line 56: The representations of related work, UCTransNet, is not unified in this article.
[2. Methods and Data]
n Figure 1: There are typos.
n Figure 2: The figure is too complex to clearly understand.
n Throughout chapter 2, there are ambiguous uses of annotations, formulas.
- For example,
- Line 96, 100: The dimension of S_i are defined differently.
- Line 96: Need to check the representation of feature size.
- Line 96: Use of operators, dot and times, are ambiguous.
- Line 116: The formulas need to be explained.
- Etc.
[3. Results]
n In chapter 3.1, causal analysis of experimental interpretation is lacking.
- Since using E1 and E3 performed better than using E1, E2, and E3 in the RRCT module, they argued that E2 should be excluded from the input because it seems to have a negative effect. However, using E2 and E3 also seems to have higher performance than using E1, E2, and E3.
- A more thorough analysis of the experiment is needed.
[4. Analysis of Experimental Results]
n In chapter 4.2.1, the relationship between the characteristics of the data set and the network models needs further explanation.
[Overall comments]
n This is a study to segment UAV images in various environments by collecting different resolution information through Transformer, ASPP, and attention module. The approach of the proposed method has high value, but it needs to be reexamined because there are many parts that are difficult to understand because the writing of the article is not smooth.
Round 2
Reviewer 1 Report
This paper has poor novelty and still have many major issues unsolved.
The algorithm is not much of novelty, the author just combine existing algorithms.
The writing and structure is hard to follow.
The authors did not provide a satifactory revision based on the above criteria.
Reviewer 2 Report
Overall, expressions in main text, figures, and formulas are not unified. (Such as subscripts, italics, etc.)
Figure 1: Concerning below, the figure needs to be reviewed or improved.
By “poling” do you mean “pooling”?
The link of arrows is unclear. Especially, the arrows linked with “C” operation in “RCCT” and “CCT” blocks are ambiguous. In “CCT” block, it looks like the arrow pointing to “+” operation on the far right is in correctly connected.
It is necessary to unify the symbols in the main text and the figure.
Etc.
The redundant use of the symbol “H” is misleading. (“Number of heads” and “Height of image”)
The ellipsis usage in equation 6 is awkward.
Description of “epo” in equation 10 is missed.
Figure 7: The label and the figure are overlapped.
Figure 9: The label is too small.
Table 4: It looks like the main text is incorrectly entered in the table.
